# Wireless Peripheral Nerve Stimulation for The Upper Limb: A Case Report

**DOI:** 10.3390/ijerph20054488

**Published:** 2023-03-03

**Authors:** Valentina Bellini, Marco Baciarello, Marco Cascella, Francesco Saturno, Christian Compagnone, Alessandro Vittori, Elena Giovanna Bignami

**Affiliations:** 1Anesthesiology, Critical Care and Pain Medicine Division, Department of Medicine and Surgery, University of Parma, Viale Gramsci 14, 43126 Parma, Italy; 2Department of Anesthesia and Critical Care, Istituto Nazionale Tumori–IRCCS, Fondazione Pascale, Via Mariano Semmola, 53, 80131 Naples, Italy; 3Departement of Anesthesia and Critical Care, ARCO ROMA, Ospedale Pediatrico Bambino Gesù IRCCS, Piazza S. Onofrio 4, 00165 Rome, Italy

**Keywords:** pain, chronic pain, neuropathic pain, nerve stimulation, peripheral nerve stimulation, implantation peripheric neurostimulation, opioid

## Abstract

Peripheral neuro-stimulation (PNS) has been proved to be effective for the treatment of neuropathic pain as well as other painful conditions. We discuss two approaches to PNS placement in the upper extremity. The first case describes a neuropathic syndrome after the traumatic amputation of the distal phalanx of the fifth digit secondary to a work accident with lack of responsiveness to a triple conservative therapy. An upper arm region approach for the PNS was chosen. The procedure had a favorable outcome; in fact, after one month the pain symptoms were absent (VAS 0) and the pharmacological therapy was suspended. The second case presented a patient affected by progressive CRPS type II in the sensory regions of the ulnar and median nerve in the hand, unresponsive to drug therapy. For this procedure, the PNS device was implanted in the forearm. Unfortunately, in this second case the migration of the catheter affected the effectiveness of the treatment. After examining the two cases in this paper, we changed our practice and suggest the implantation of PNS for radial, median and/or ulnar nerve stimulation in the upper arm region, which has significant advantages over the forearm region.

## 1. Introduction

Peripheral nerve stimulation (PNS) has been proved effective for treating neuropathic pain, as well as other painful conditions not responding to optimal medical management [1,2,3], including chronic pain syndromes such as post-traumatic and post-surgical neuropathy, occipital neuralgia and complex regional pain syndromes, or neurological pathologies such as migraines, daily headaches or cluster headaches. Lesions or disorders of the peripheral nervous system generally present a unique challenge to the treating physician because the associated neuropathic pain can be extremely resistant to typical pain treatments. A significant percentage of patients present resistance to treatment or a high number of side effects that affect the quality of everyday life [1].

Peripheral nerve field stimulation (PNFS) is a valid treatment for neuropathic pain affecting the torso, the abdominal region and for fibromyalgia [4,5,6,7,8,9]. Peripheral nerve and field stimulation is a novel type of neuromodulation. It represents a minor surgical procedure that implants electrodes into the body to change the way the peripheral nervous system works. The stimulation is performed on peripheral nerves and the field consists of placing the electrodes directly on the nerves or under the skin in the region of pain [4].

It is also effective in treating peripheral neuralgia, refractory migraine, cranio-facial neuropathic pain and cluster headaches [10,11]. Additionally, in complex regional pain syndrome (CRPS), peripheral or spinal cord stimulators (SCS) are effective options for pain management [12]. This has been used with positive success to treat pain syndromes after surgery. Hassembush et al. reported a moderate reduction of pain (63%) in 30 patients with CRPS after the implantation of PNS; pain reduction was sustained through at least two years of follow-up [12,13]. The most accredited theories on the functioning of peripheral nerve stimulation block pain through a combination of central and peripheral mechanisms.

Melzack and Wall demonstrated that stimulation of afferent axons (Aβ fibers) that transmit touch, vibration, and proprioception signals simultaneously inhibits nociceptive inputs from peripheral Aδ fibers and C fibers, which send pain signals. Inhibitory interneurons will inhibit the transmission of pain signals to higher central nervous system centers. The repetitive stimulation of peripheral nerves will increase thresholds for Aδ fibers and C fibers and reduce arousal [14]. 

The advantages of PNS, as compared to SCS, are the less invasive implantation procedure, direct access to the anatomical target, faster recovery from surgery, and surgery-related costs. Disadvantages include a higher risk of migration, catheter fracture and specific, smaller areas of effect (dependent on the stimulated nerve(s)). Most recent PNS models consist of an implantable wire with stimulating contacts on one end, and an electromagnetic coil receiver on the other end. The middle portion of the lead contains circuitry capable of changing the frequency, intensity, and timing of the electrical stimuli. Energy for the stimulation comes from wearable devices acting both as induction generator and stimulator controllers. At the moment, the available scientific evidence is limited, the techniques used are heterogeneous, and the indications are dissimilar among the different authors. However, the evidence of benefits in selected cases where other treatments have failed is highly significant. Concerning the upper extremities, the difficulties increase given the complex anatomical nature that includes using the hands for everyday tasks.

The aim of this paper is to describe two useful, alternative anatomical approaches for PNS implantation, in the medial aspect of the upper arm and at the anterior aspect of the forearm, detailing pros and cons of either approach. Written consent has been obtained from patients for the publication of these case reports in standard institutional forms. The manuscript was written following the CARE guidelines. All subjects gave their informed consent for inclusion in these case reports, according to our institutional criteria. Our center does not require specific approval by the Ethics Committee for case reports.

## 2. Case Description

### 2.1. First Patient

A thirty year-old man presented to our center complaining of functional limitation and painful neuropathic symptoms at the ulnar aspect of the fifth digit of his right hand, with irradiation along the forearm. The pain began after the traumatic amputation of the distal phalanx of the fifth digit secondary to a work accident. Due to a lack of responsiveness to conservative therapy with non-steroidal anti-inflammatory drugs (NSAIDs) and opioids (pain evaluated with Numerical Rating Score (NRS) 8 in the daytime and NRS 7 at nighttime), the patient underwent the positioning of a PNS. Chronic treatment with pregabalin 150 mg twice daily, tramadol 100 mg twice daily and amitriptyline 10 mg in the evening was prescribed. The patient described her pain as a constant 8/10 daily, with a poor quality of life (QoL). One week before surgery, a test block of the ulnar nerve with 1% lidocaine using an ultrasound-guided technique was performed. Immediately after the procedure, the block produced analgesia and relieved the pain (NRS 0). Therefore, in agreement with the patient, it was decided to implant a wireless peripheral stimulator.

The patient was positioned supine, with the arm abducted at 90 degrees and the elbow flexed. The path of the ulnar nerve was identified using a high-frequency linear probe at the level of the cubital fossa. The procedure was performed with light sedation with midazolam and fentanyl, and local anesthesia with 2% lidocaine. The nerve path was followed towards the axilla and the optimal length of the lead/radiofrequency receiver was estimated.

Afterwards, a small incision in the antero-medial side of the arm was performed, a few centimeters distant from the elbow crease. The quadripolar catheter was inserted into this surgical site along the course of the right ulnar nerve. Due to the relative complexity of the surgical procedure, two operators performed the placement of the catheter: the first one maintained adequate ultrasound visualization of the ulnar nerve, and the second one directed the introducer. Paresthesia was used to verify the adequacy of the placement, indicated for the adequate reduction or absence of pain sensation. The technology chosen was the implantable quadripolar neurostimulation system, with integrated wireless technology. “StimQ” is a system for the stimulation of the peripheral nervous system, consisting of quadripolar leads with an integrated stimulator. The system is able to stimulate in various therapeutic modalities such as tonic stimulation, burst stimulation (StimSurge), and high frequency stimulation (up to 1499 Hz).

Over the following days, the patient presented edema in the upper arm region that was treated by applying ice until the resolution of this symptom (postoperative day 5). After 1 week, NRS reduced from 8 to 4 with a de-escalation in pain treatment (pregabalin 75 mg twice daily, tramadol 50 mg in the morning and 50 mg in the evening and amitriptyline 5 mg in the evening). During the following checks, the PNS appeared to be working perfectly (post-implantation NRS was zero at rest and incident).

After one month, the patient was evaluated again at our center; the pain symptoms were absent (VAS 0), and the pharmacological therapy was suspended. One year later, the device is fully functional and the patient is not taking any drugs for neuropathic pain.

### 2.2. Second Patient

A fifty-year-old man presented with progressive CRPS type II affecting the sensory regions of the ulnar and median nerve in the hand. Symptoms included swelling, stiffness, dyschromia, hypoesthesia and burning pain. The patient’s condition was a consequence of vehicular trauma, with a complex dislocation of his right third finger associated with multiple tendon lesions. Our patient had undergone multiple hand surgeries and was no longer responsive to opioid and nonsteroidal anti-inflammatory drugs (NRS 8 at rest and incident). The prescribed drug therapy included pregabalin 600 mg twice daily, tapentadol 100 mg twice daily, amitriptyline 10 mg in the evening, and requiring rescue therapy with NSAIDs at least three times a week. According to the recommendations, one week before the procedure an ultrasound-guided block test on the ulnar and median nerves with 1% lidocaine was performed. After about 3 minutes, the patient reported complete resolution of pain symptoms (NRS 0). Therefore, a PNS trial was proposed. 

For surgery, the patient was positioned supine with the operative arm abducted and slight dorsal flexion of the wrist. The ulnar and median nerve were identified with ultrasound in the cubital fossa and in the elbow crease, respectively. Additionally, in this case, mild sedation with anxiolytics and opiates was performed, in addition to local anesthesia, without major difficulties.

The PNS trocar was inserted along the median nerve in the proximal-to-distal di-rection under continuous ultrasound guidance. The painful area was detected by using stimulation paresthesia within the distribution of the target nerve root. 

The same procedure was repeated for the right ulnar nerve (Figure 1).

Despite the correct functioning of electro-catheters, over the second postoperative day a surgical review of the implantation site was performed for the excessive print of catheters on the skin.

After a week, NRS reduced from 8 to 5, with a halving of the drug therapy (pregabalin 300 mg twice daily, tapentadol 50 mg twice daily, amitriptyline 5 mg in the evening).

However, after 15 days, the patient contacted the center due to the recovery of neuropathic pain in the territory of the median nerve (NRS 8); it was decided to restore the dosage of the pre-intervention pharmacological therapy. At an X-ray check, the catheter in the median nerve area appeared to have migrated. It was therefore decided to proceed with the explantation of the device.

## 3. Discussion

The analysis of these two patients, treated with the same technique but with two different approaches, similar from a procedural point of view but different considering the anatomical areas, allows us to describe, in objective terms, the pros and cons of both the solutions here described.

There is an evident difficulty in identifying the nerve pathways in the upper arm region. Undoubtedly, this problem can be partially solved using the two-operator technique. Other positive aspects of the upper arm region are the reduced presence of anatomical structures and the vast implantation region.

In the forearm, there are eight muscles, tendons and ligaments. Instead, in the upper arm region area there are only three muscles.

Otherwise, the choice to proceed in the forearm region was technically easy. Among the disadvantages, we met a significant number of complications; post-operative pain, and increased risk of migration, due, probably, to the presence of a considerable number of anatomical structures that cause a reduction in the space available for the device.

The literature and the operating instructions provided by the manufacturer do not clearly indicate, so far, the best site and modalities for the implantation of a PNS.

We suggest always performing an ultra-sound check before the procedure to visualize the path of ulnar, median and radial nerves.

Complications with implantable devices include infection (4%) and skin lesions (2%). Due to the anatomical characteristics of the upper limbs, device migration is always a concern (15%). Percutaneous techniques performed with ultrasound guidance theoretically have more lead migration than open techniques because the device is not secured to the underlying muscle fascia, and older leads did not have integrated anchoring technology. The variation of the method discussed in this clinical case could reduce this incidence [15]. 

There have yet to be data from comparative studies that have evaluated these techniques.

Older PNS devices used split-ring electrodes that circumferentially wrapped the nerve, increasing the rates of nerve entanglement and perineural fibrosis. Modern electrodes are designed to avoid this problem. However, there are other failures in the device, such as cable breakage (11%). In the authors’ opinion, the variation of the technique presented could reduce this incidence.

Data on the longevity of these implants are currently unavailable. However, there are concerns that migration may occur over time, limiting the device’s efficacy, and would be unrelated to the technique used. Similarly, a failure to adequately treat pain is most commonly due to device migration. Additionally, this cause would be deeply associated with the method used.

Ishizuka et al. retrospectively reviewed why patients required reoperation after initially successful PNS and found that 64% of patients needed one or more additional surgeries; lead migration was the most common cause (33%) of device failure [16].

The procedures performed on cadavers showed a superior margin of safety by using the upper arm approach.

## 4. Conclusions

In conclusion, after examining the two cases portrayed in this paper, it is possible to suggest that the implantation of PNS to stimulate the radial, median and ulnar nerves through the upper arm region has benefits in comparison to the forearm region. Although the technique results are more complex, the presence of a small number of anatomical structures in the antero-medial side of the arm should minimize the risk of the main post-operative complications. Spinal cord stimulators have been used for decades for the non-surgical treatment of chronic low back pain, with demonstrable benefit. In contrast, peripheral nerve stimulators have been comparatively little used. The literature on the upper extremities contains little evidence, and more randomized studies are needed to prove our hypothesis.

## Figures and Tables

**Figure 1 ijerph-20-04488-f001:**
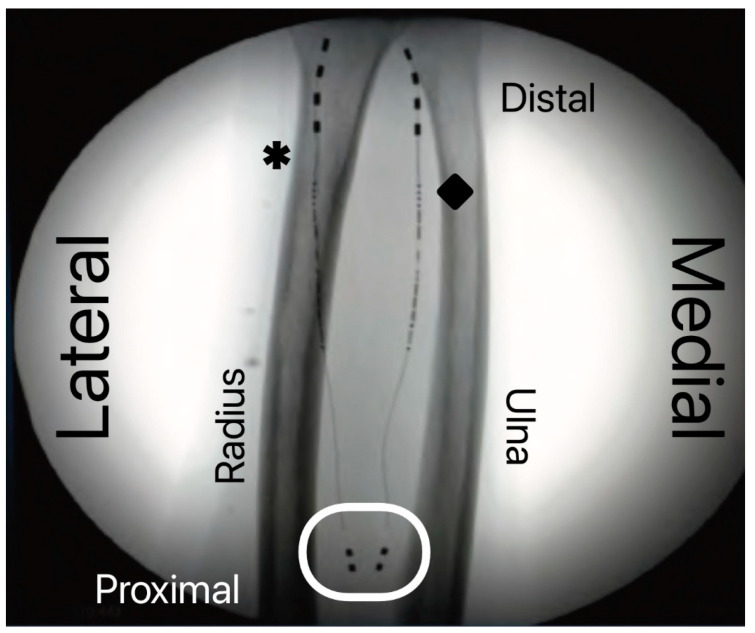
Fluoroscopy of the forearm illustrating the final position of the two peripheral nerve stimulation leads. The tips of the leads, with four stimulation contacts, lie next to the radial (asterisk) and median (diamond) nerve just proximal to the radial epiphysis. The white ellipse highlights the leads’ antennae which receive electrical energy from the external generator; energy is then converted into the actual stimulation current by the circuitry embedded in the body of the lead.

## Data Availability

The data can be requested from the corresponding author for a reasonable purpose.

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
