# Peer review of "Wireless Peripheral Nerve Stimulation for The Upper Limb: A Case Report"

_ijerph, 2023, doi:10.3390/ijerph20054488_

Round 1
Reviewer 1 Report
First of all, I would like to express my gratitude for the opportunity to review this research. I agree with the authors about the information contained in the main text. However, a few considerations arise for me, which I leave raised below:
General Consideration
I would like to emphasize the importance of the study for pain. In spite of the interest shown during the reading of this investigation, maybe the design of the present study may not be sufficient for publication in this journal. The reason for this is the high-quality journal. But at the same time, we think that the authors could improve it. It will probably be interesting to reanalyse this work in the future. Therefore, I encourage the authors to restructure the information according to Care Checklist (Microsoft Word - CAREchecklist-English.docx (squarespace.com)).
Specific Consideration
Key Words: 5 of the 7 words do not correspond to MESH TERMS
Abstract: Main Symptoms, diagnosis and clinical findings should be included in this section.
Introduction: 8/13 references include are more than 5 years old.
Patient Information: Both the diagnosis and the examination of the patient should be specified, as well as the outcomes obtained of the treatment (patient specific information). To this aim, it would be interesting to know about previous treatments which have been applied to patients. With respect to the treatment applied, it should be described the dosage and not only location of the electrode and the surgical procedure.
As for a therapeutic intervention is concerned (VAS), we believe it would be appropriate to add changes in therapeutic intervention (other outcomes), when treating neuropathic pain. In this regard, follow-up and outcomes (patient assessed outcomes, adherence y tolerability and follow-up results may be showed. Ç
For its part, the article should include a discussion section with references. Furthermore, the authors should be taken into account strengths and limitations, and they also should include the assessment of possible causes of the conclusions.
Finally, it is understood that the perspective of the patient and clinical implications for clinicians should also be considered.
To sum up and from my humble point of view, the answers to these considerations could be included in the text because of their great relevance in order to enhance the research.
Author Response
To the attention of:
Prof. Dr. Paul B. Tchounwou,
Department of Biology, College of Science, Engineering and Technology, Jackson State University, 1400 Lynch Street, Box 18750, Jackson, MS 39217, US
Parma,February25, 2023
Dear Editor,
We appreciate the thorough review of our manuscript and the helpful comments provided by all the reviewers. We have tried to address all of the issues that were brought to our attention. The changes to the revised manuscript are marked in red. Please find a our reply of each comment:
Reviewer 1
First of all, I would like to express my gratitude for the opportunity to review this research. I agree with the
authors about the information contained in the main text. However, a few considerations arise for me,
which I leave raised below:
We would like to thank Reviewer 1 for his time and appreciation for our work.
General Consideration
I would like to emphasize the importance of the study for pain. In spite of the interest shown during the
reading of this investigation, maybe the design of the present study may not be sufficient for publication in
this journal. The reason for this is the high-quality journal. But at the same time, we think that the authors
could improve it. It will probably be interesting to reanalyse this work in the future. Therefore, I encourage
the authors to restructure the information according to Care Checklist (Microsoft Word - CAREchecklist-
English.docx (squarespace.com)).
We would like to thank reviewer 1 for his valuable suggestion, very fair. We followed his directions.
Specific Consideration
Key Words: 5 of the 7 words do not correspond to MESH TERMS
We thank you and corrected.
Abstract: Main Symptoms, diagnosis and clinical findings should be included in this section.
We agreed and corrected.
Introduction: 8/13 references include are more than 5 years old.
Thank you. We have included some more recent reference
Patient Information: Both the diagnosis and the examination of the patient should be specified, as well as
the outcomes obtained of the treatment (patient specific information). To this aim, it would be interesting
to know about previous treatments which have been applied to patients. With respect to the treatment
applied, it should be described the dosage and not only location of the electrode and the surgical
procedure.
We agreed and corrected.
As for a therapeutic intervention is concerned (VAS), we believe it would be appropriate to add changes in
therapeutic intervention (other outcomes), when treating neuropathic pain. In this regard, follow-up and
outcomes (patient assessed outcomes, adherence y tolerability and follow-up results may be showed. Ç
We agreed and corrected
For its part, the article should include a discussion section with references. Furthermore, the authors
should be taken into account strengths and limitations, and they also should include the assessment of
possible causes of the conclusions.
We agreed and corrected
Finally, it is understood that the perspective of the patient and clinical implications for clinicians should also
be considered.
We agreed and corrected
To sum up and from my humble point of view, the answers to these considerations could be included in the
text because of their great relevance in order to enhance the research.
We agreed and corrected
In conclusion, we hope to have addressed all of the issues raised by your thorough reviews. We would be glad to further modify our work upon your request.
On behalf of the other authors, I would like to extend my gratitude for your assistance and consideration.
I look forward to your response.
Kind regards,
Elena Giovanna Bignami

Reviewer 2 Report
Overall, this is an interesting article that described two approaches for peripheral neurostimulation (PNS) implantation.
The authors compared the PNS device implantation in the upper arm versus the forearm. They conclude that the upper arm region has benefits in comparison to the forearm region, based on PNS device implantation in the forearm had complications such as post-operative pain and migration.
Major concerns:
1. It was not added IRB # approval. The journal suggests the following example: "All subjects gave their informed consent for inclusion before they participated in the study. The study was conducted in accordance with the Declaration of Helsinki, and the protocol was approved by the Ethics Committee of XXX (Project identification code)."
2. The MS needs a better description of the criteria used to compare both sites of the PNS device implantation (i.e pain, complications).
3. I suggest labeling figure 1 to be easiest for the reader to identify the elements in the figure.
Minor concerns:
1. Add comma lines 96,101,106.
Author Response
To the attention of:
Prof. Dr. Paul B. Tchounwou,
Department of Biology, College of Science, Engineering and Technology, Jackson State University, 1400 Lynch Street, Box 18750, Jackson, MS 39217, US
Parma,February25, 2023
Dear Editor,
We appreciate the thorough review of our manuscript and the helpful comments provided by all the reviewers. We have tried to address all of the issues that were brought to our attention. The changes to the revised manuscript are marked in red. Please find a our reply of each comment:
Reviewer 2
Overall, this is an interesting article that described two approaches for peripheral neurostimulation (PNS)
implantation.
The authors compared the PNS device implantation in the upper arm versus the forearm. They conclude
that the upper arm region has benefits in comparison to the forearm region, based on PNS device
implantation in the forearm had complications such as post-operative pain and migration.
We thank reviewer 2 for the time he has dedicated to us and the valuable suggestions he has been able to
provide us to improve the quality of the paper.
Major concerns:
- It was not added IRB # approval. The journal suggests the following example: "All subjects gave their
informed consent for inclusion before they participated in the study. The study was conducted in
accordance with the Declaration of Helsinki, and the protocol was approved by the Ethics Committee of
XXX (Project identification code)."
We thank the reviewer for his attention to ethical issues. We corrected by indicating the rules we followed,
in full coherence with the ethical issues raised.
- The MS needs a better description of the criteria used to compare both sites of the PNS device
implantation (i.e pain, complications).
We agreed and corrected
- I suggest labeling figure 1 to be easiest for the reader to identify the elements in the figure.
We agreed and corrected
Minor concerns:
- Add comma lines 96,101,106.
We agreed and corrected
In conclusion, we hope to have addressed all of the issues raised by your thorough reviews. We would be glad to further modify our work upon your request.
On behalf of the other authors, I would like to extend my gratitude for your assistance and consideration.
I look forward to your response.
Kind regards,
Elena Giovanna Bignami

Round 2
Reviewer 1 Report
I would like to thank the authors for their efforts. Thank you for your work.